# The Chromatin-Oxygen Sensor Gene *KDM5C* Associates with Novel Hypoxia-Related Signatures in Glioblastoma Multiforme

**DOI:** 10.3390/ijms231810250

**Published:** 2022-09-06

**Authors:** Denise Drongitis, Lucia Verrillo, Pasqualino De Marinis, Pasquale Orabona, Agnese Caiola, Giacinto Turitto, Alessandra Alfieri, Sara Bruscella, Marisa Gentile, Vania Moriello, Ettore Sannino, Ines Di Muccio, Valerio Costa, Maria Giuseppina Miano, Alberto de Bellis

**Affiliations:** 1Institute of Genetics and Biophysics Adriano Buzzati-Traverso, CNR, 80131 Naples, Italy; 2Maria Rosaria Maglione Foundation Onlus, 80122 Naples, Italy; 3A.O.R.N. S. Anna and S. Sebastiano Hospital, Division of Neurosurgery, 81100 Caserta, Italy; 4A.O.R.N. S. Anna and S. Sebastiano Hospital, Division of Pathology, 81100 Caserta, Italy; 5A.O.R.N. S. Anna and S. Sebastiano Hospital, Division of Oncology, 81100 Caserta, Italy

**Keywords:** glioblastoma multiforme, KDM5C, 5-aminolevulinic acid fluorescence-guided surgery (5-ALA FGS), HIF1A-KDM5C axis, GBM with epilepsy, hypoxic microenvironment

## Abstract

Glioblastoma multiforme (GBM) is a fatal brain tumor without effective drug treatment. In this study, we highlight, for the first time, the contribution of chromatin remodeling gene Lysine (K)-specific demethylase 5C (*KDM5C*) in GBM via an extensive analysis of clinical, expression, and functional data, integrated with publicly available omic datasets. The expression analysis on GBM samples (N = 37) revealed two informative subtypes, namely KDM5C^High^ and KDM5C^Low^, displaying higher/lower KDM5C levels compared to the controls. The former subtype displays a strong downregulation of brain-derived neurotrophic factor (*BDNF*)—a negative KDM5C target—and a robust overexpression of hypoxia-inducible transcription factor-1A (*HIF1A*) gene, a *KDM5C* modulator. Additionally, a significant co-expression among the prognostic markers *HIF1A*, *Survivin*, and *p75* was observed. These results, corroborated by *KDM5C* overexpression and hypoxia-related functional assays in T98G cells, suggest a role for the HIF1A-KDM5C axis in the hypoxic response in this tumor. Interestingly, fluorescence-guided surgery on GBM sections further revealed higher *KDM5C* and *HIF1A* levels in the tumor rim niche compared to the adjacent tumor margin, indicating a regionally restricted hyperactivity of this regulatory axis. Analyzing the TCGA expression and methylation data, we found methylation changes between the subtypes in the genes, accounting for the hypoxia response, stem cell differentiation, and inflammation. High *NANOG* and *IL6* levels highlight a distinctive stem cell-like and proinflammatory signature in the KDM5C^High^ subgroup and GBM niches. Taken together, our results indicate HIF1A-KDM5C as a new, relevant cancer axis in GBM, opening a new, interesting field of investigation based on *KDM5C* as a potential therapeutic target of the hypoxic microenvironment in GBM.

## 1. Introduction

Glioblastoma multiforme (GBM) is one of the most aggressive and damaging tumors of the brain without effective targeted chemotherapeutic agents [1]. GBM is characterized by high heterogeneity and poor survival with a median rate of 12–16 months after diagnosis [2]. It is a fast-growing tumor characterized by the presence of oxygen deficiency (hypoxia) with central necrosis, robust angiogenesis, intense resistance to apoptosis, and genomic instability [3]. Currently, patients with GBM undergo surgical removal of the tumor mass and then radiotherapy and/or chemotherapy using temozolomide (TMZ) protocols that, however, are still completely inadequate to effectively combat GBM [2,4]. To date, the poor understanding of the molecular mechanisms underlying GBM aggression, multiple drug resistance (MDR), and relapse have prevented the development of effective targeted therapies [5]. At the basis of the aforementioned processes, several studies have proved the critical role played by hypoxia characterizing the tumor microenvironment (TME), but little is known about the molecular players involved and how to manipulate them.

The frequent occurrence of somatic alterations in the genes responsible for chromatin remodeling and transcriptional control in cancer is the Achilles’ heel of tumors, which allows the identification of epigenetic biomarkers as new possible drug targets [6,7,8]. In this regard, it has definitely established the key role of Jumonji-C (JmjC) histone demethylases (KDMs) as chromatin oxygen sensors required for the cellular response to hypoxia [9]. Mechanistically, JmjC KDMs are 2-oxoglutarate-dependent dioxygenases whose enzymatic activity requires oxygen and Fe^2+^ to promote the hydroxylation reaction necessary for the removal of the methyl groups [9]. Among them, X-chromosome Lysine-specific demethylase 5C (*KDM5C*) is a JmjC gene involved in *p53* gene expression regulation, having a significant role in tumor cell proliferation, migration, and drug resistance [10]. The KDM5C protein catalyzes H3K4me3-me2 demethylation [11,12], and depending on the methylation site, it can either activate or repress gene transcription [13]. Somatic mutations in *KDM5C* were identified in renal cell carcinoma [14], pancreatic cancer [15], and in primary and secondary chemoresistant pediatric acute myeloid leukemia [16]. In addition to mutations, alterations in the *KDM5C* expression levels have been related to the development of various cancers. Indeed, *KDM5C* overexpression is associated with increased tumor cell proliferation and tumorigenic progression in multiple cancer types, including colorectal [17], breast [18], ovary [19], and prostate [20]. Conversely, *KDM5C* downregulation is associated with genomic instability in clear cell renal cell carcinoma (ccRCC) [14].

The role of KDM5C in cancer is still controversial, given the dual -pro-oncogenic and suppressive properties, which are highly tumor-specific. KDM5C alterations have been proposed as a negative prognostic marker in various cancers and have been also used to predict survival benefits upon immune checkpoint inhibitor treatments [21]. In ccRCC, KDM5C specifically regulates the expression of several hypoxia-inducible factor (Hif)-related genes [22], and its deficiency promotes tumorigenicity by reprogramming glycogen metabolism and inhibiting ferroptosis [23]. Most importantly, pharmacological KDM5C inhibition blocks cell growth and tumorigenesis [24] by repressing oncogenic target genes [25] or, alternatively, by counteracting genomic instability [14].

*KDM5C* was initially identified as an X-linked Intellectual Disability (XLID) gene with a critical role for brain development and functioning, as evident by the discovery of hereditary and de novo mutations in patients with neurodevelopmental diseases (NDDs), presenting intellectual disability (ID), epilepsy, and autism [26]. Several independent studies [27,28], including ours [29,30,31], reported *KDM5C* as widely expressed in both neurons and astrocyte cells and its activity as required for neuronal maturation and plasticity.

Here, we have explored for the first time the potential role of *KDM5C* in the pathophysiology of GBM. We analyzed the expression of this chromatin oxygen sensor gene in GBM tissues isolated both from conventional surgery and the most recent 5-aminolevulinic acid fluorescence-guided surgery (5-ALA FGS) method. This analysis allowed us stratifying GBM patients into two subgroups (KDM5C^Low^ and KDM5C^High^) in which the expression of KDM5C targets and hypoxic-related markers was extensively studied. Additionally, taking advantage of public omic datasets, we disclosed differences in gene expression and methylation between the tumor subgroups, identifying relevant altered and interconnected processes, such as the hypoxia response, inflammation, and stemness. Overall, our results highlight the role of KDM5C in GBM, identifying the new hypoxia-induced signature exclusively localized in the tumor tissue.

## 2. Results

### 2.1. Stratification of GBM Patients Based on KDM5C Gene Expression Levels

Considering the debated role of KDM5C as an oncogene or tumor suppressor and its crucial contribution in brain functioning, we first aimed to establish *KDM5C* expression levels by real time PCR (RT-PCR) in tumor samples (N = 37) derived from GBM patients compared to brain control samples (N = 8). Analyzing the entire cohort of GBM samples (KDM5C^Tot^), whose clinical features are summarized in Table 1, we found no significant differences in *KDM5C* expression compared to the control samples (Appendix A). However, as we noted highly heterogeneous *KDM5C* levels across the sample cohorts, we sought to stratify them into two subgroups—KDM5C^Low^ and KDM5C^High^ (Figure 1A)—according to the *KDM5C* expression. As expected, a strong increase in the amount of KDM5C protein was evident in the protein lysates obtained from biopsies of the KDM5C^High^ GBM subgroup (Figure 1B), whereas no detectable band was visible in the WB assay from the samples of the KDM5C^Low^ GBM subgroup. Looking for a clinical feature associated with KDM5C changes, we analyzed the baseline patient characteristics of each cohort. Despite how the KDM5C levels do not significantly associate either with clinicopathological characteristics (Appendix A) or survival (Appendix A) in the two GBM subgroups, significant changes in Ki67 and p53 protein expressions were observed in KDM5C^Low^ patients with (Epilepsy+) compared to those without epilepsy (Epilepsy−) and in KDM5C^Low^ patients with (IDH^mut^) or without (IDH^WT^) *IDH1/2* mutation (Figure 1C), respectively. These findings indicate a possible link between KDM5C, and these two prognostic markers widely used as poor prognosis indicators in GBM patients [32]. Next, we explored whether low and high *KDM5C* levels might impact the expression of the negative KDM5C-responsive gene brain-derived neurotrophic factor (*BDNF*), implicated in GBM pathogenesis [33]. As expected, by analyzing the entire GBM cohort (KDM5C^Tot^), no significant differences in *BDNF* expression were found between the two groups and compared to the control samples (Appendix A). However, a significant *BDNF* decrease was observed specifically in the *KDM5C*^High^ subgroup compared to the control samples (Figure 1D); no significant variation was observed in the *KDM5C*^Low^ tumor samples that displayed expression values comparable to the healthy tissues (Figure 1D). Overall, these results suggest the potential use of KDM5C as a molecular stratification factor in GBM, with a particular attention for patients with epilepsy and *IDH1/2* mutations.

### 2.2. Hypoxia-Related Signatures Correlate with KDM5C Levels

Given the role of the KDM5C protein as an oxygen sensor [9], we further analyzed the expression levels of hypoxic markers aiming to identify the tumor signatures associated with the changes in *KDM5C*. Noteworthy, hypoxia, a characteristic of almost all types of solid tumors, has been associated with a poor outcome in GBM [34]. We therefore measured the expression of hypoxia-inducible factor 1-alpha, encoded by the *HIF1A* gene reported to be a direct modulator of *KDM5C* expression [22,35] and to play a critical role in GBM progression [36] and temozolomide responsiveness [37]. In line with its pro-tumorigenic feature, *HIF1A* is overall induced in GBM samples (vs. the control), reaching its highest expression levels in the KDM5C^High^ subgroup (Figure 1D). Despite a trend of over-expression also observed in the KDM5C^Low^ subgroup, it did not reach the threshold (*p* > 0.05) of statistical significance (Figure 1D). Since high levels of HIF-1α are strongly correlated with the expression of survivin, a protein involved in tumor progression [38], we analyzed the transcript levels of the corresponding gene Baculoviral IAP Repeat-Containing Protein 5 (*BIRC5)*. Interestingly, in both tumor subgroups, we found a marked *BIRC5* overexpression compared to the control samples (Figure 1D), even though a very strong and significant positive correlation between *HIF1A* and *BIRC5* was observed only in the KDM5C^High^ subgroup (Figure 1E and Appendix A).

Afterwards, we analyzed the expression of the p75 neurotrophin receptor (*p75*^NTR^) that, as previously reported by Tong et al. [39], positively regulates the level of *HIF1A* and correlates with hypoxia-induced stemness in glioma. We measured an increased expression of *p75*^NTR^ and a positive correlation with *HIF1A*, which is significant only in the *KDM5C*^High^ subgroup vs. the control samples (Figure 1F,G). We also tested the transcript level of the nerve growth factor (*NGF*) that, through the binding to receptor p75^NTR^, is involved in glioma proliferation [40]. No significant variation in the *NGF* levels were observed in the entire cohort of tumor samples or in the two subgroups compared to the controls (Appendix A). Collectively, despite the limitations due to the low number of patients, these findings indicate that GBM tumors expressing higher *KDM5C* levels are positively associated with increased *HIF1A*, suggesting a potential pro-tumorigenic activity of *KDM5C* in GBM. On the opposite side, the marked repression of *BDNF* in the *KDM5C*^High^ subgroup suggests a possible direct role of KDM5C in the promoter’s demethylation in these tumor samples.

### 2.3. Hif-1α Stabilization Induces KDM5C Increase and BDNF Repression in T98G Glioblastoma Cell Line

To functionally investigate the impact of *KDM5C* overexpression on the *BDNF* and *HIF1A* levels, we transiently transfected the T98G glioblastoma cell line with *KDM5C* plasmid (see Materials and Methods). After assessing the transfection efficiency, we measured the expression of *BDNF*, *HIF1A*, *NGF*, and *p75*^NTR^ mRNAs in transfected cells. As shown in Figure 2A, *KDM5C*-overexpressing cells display a significant downregulation of *BDNF*; while no effects were observed for *HIF1A* (Figure 2A), *NGF*, and *p75*^NTR^ (Appendix A), suggesting that one of the primary events of *KDM5C* induction in tumor cells could be the transcriptional repression of the *BDNF* gene. However, considering that hypoxia—through the involvement of Hif-1α—has been reported to regulate histone demethylases, including *KDM5C* [35], we used cobalt chloride (CoCl_2_), a hypoxia-mimicking chemical [41], to stabilize Hif-1α in T98G cells. Interestingly, *HIF1A* mRNA increasingly correlates with high levels of *KDM5C* mRNA (Figure 2B). Furthermore, in line with *KDM5C* over-expression, CoCl_2_-treated tumor cells also display *BDNF* repression and *p75*^NTR^ upregulation (Figure 2B). It should be noted that, in the CoCl_2_ conditions employed, in line with the published data [42], the growth or survival rate of T98G cells were unchanged. Overall, the in vitro data on the GBM tumor cell line indicate that, in hypoxic conditions, high levels of *p75^NTR^* may correlate with high levels of *HIF1A* and *KDM5C*, a negative regulator of *BDNF*.

### 2.4. Regional Heterogeneity of KDM5C and HIF1A Expression Profiles in Distinct GBM Areas Isolated by 5-Aminolevulinic acid Fluorescence-Guided Surgery

Since GBM contains several tumor microenvironments (TMEs) with irregular distributions of signaling networks [43], we wondered whether the gene expression levels of *KDM5C*, *BDNF*, and *HIF1A* differed in distinct GBM tumor areas. Taking advantage of 5-aminolevulinic acid (5-ALA) fluorescence-guided surgery (FGS), we isolated the central region of the tumor, the infiltrating front, and the healthy surrounding tissue, respectively, characterized by bright (5-ALA++; intense), weak (5-ALA+; moderate), and no fluorescence (5-ALA-; Figure 3A and Appendix A) [44]. In terms of histological characteristics, these areas correspond to three dominant TME niches: 5-ALA++ corresponds to the tumor core (TC) characterized by severe hypoxia, 5-ALA+ corresponds to the tumor rim (TR) with infiltrating glioma cells characterized by high mitotic activity, and 5-ALA- corresponds to the tumor margin (TM) considered the intersection between infiltrating tumor cells and nontumoral brain tissue (Figure 3A). The expression analysis was carried out in representative regions displaying strong (5-ALA++), moderate (5-ALA+), or no 5-ALA (5-ALA-) fluorescence obtained from seven GBM patients undergoing 5-ALA FGS. Interestingly, we measured higher *KDM5C* expression levels in the TR region compared to the TM region (*p*-value 0.004). Moreover, significantly higher levels of *HIF1A* were measured in both TC and TR compared to the TM area (Figure 3B and Appendix A). No significant differences were found for the *BDNF* levels among the three areas, and there is a higher level of *HIF1A* in both TC and TR than in TM (Figure 3B). Furthermore, *p75*^NTR^ expression was significantly increased in TC and TR compared to the TM (Figure 3B). We also measured higher *NGF* levels in TR (vs. TC and TM; Figure 3B). Overall, the results of the expression analysis based on the 5-ALA FGS approach indicate that the tumor margins are characterized by low levels of *KDM5C*, *HIF1A*, *p75*^NTR^, and *NGF*, whereas the tumor border regions (TR) are significantly enriched for these markers, and, in line with the highly hypoxic features of the tumor core, it is mainly characterized by increased *HIF1A* and *p75*^NTR^ expression.

### 2.5. GBMs Expressing High or Low KDM5C Display Markedly Different Expression and Methylation Patterns

Prompted by the evidence of the highly variable expression of *KDM5C* in GBM patients and hypothesizing that such heterogeneity might result in distinct tumor subtypes having peculiar tumorigenic features, we took advantage of public expression and methylation data available from The Cancer Genome Atlas (TCGA; GBM cohort). As the expression data from healthy controls for GBM are missing in TCGA, we used RNA-Seq data from post-mortem brains in GTeX to verify that *KDM5C* is indeed overexpressed in tumor samples (Figure 4A). Then, using cBioportal, we created two subgroups of samples according to the *KDM5C* expression levels (see Appendix A). No differences in the clinical features, including the probability of overall survival (Appendix A), were found between the two subgroups. Moreover, differential mRNA expression and methylation were assessed between them (Figure 4B,C). As evident in Figure 4B,C (left panels), the two tumor subgroups largely differ in terms of gene expression and methylation. Interestingly, Gene Ontology and the pathway analysis on differentially expressed genes revealed the enrichment of inflammatory-related, as well as of chromatin remodeling, processes (Figure 4B, right panel). Similarly, the analysis of differentially methylated genes indicated “stem cell differentiation”, “hypoxia”, “p53 signaling”, and “inflammation” as the most relevant processes and/or pathway affected (Figure 4C, right panel).

Finally, classifying the genes altered between KDM5C^Low^ and KDM5C^High^ according to the consistency between the methylation and expression values, we focused on the two clusters of genes characterized by hypomethylation and overexpression (Figure 4D, left panel), as well as by hypermethylation and downregulation (Figure 4D, right panel). In particular, using this approach, we identified 117 genes hypomethylated (and overexpressed) in KDM5C^High^ tumor samples and only five genes showing the opposite trend (Figure 4D). Interestingly, using the Panther classification system (over-representation test), we observed that the genes consistently regulated by methylation changes belong to “RNA processing/splicing”, “chromatin remodeling”, and the “cell cycle” (Figure 4E). Moreover, when analyzing the potential interactions occurring among the related proteins (by STRING), we found an interesting network, including 51 interacting proteins among these gene products, as highlighted in Figure 4F, with significant enrichment in *Chromatin organization* (GO:0006325), *Regulation of transcription* (GO:0045944), *RNA splicing* (GO:0043484), *RNA processing* (GO:0006396), *Mitotic cell cycle* (GO:0000278), and *Regulation of cell cycle* (GO:0010564). Among these genes, 90% (46/51) are implicated in cancer (Appendix A), including seven involved in glioblastoma pathogenesis, such as Polypyrimidine tract binding protein 2 (*PTBP2*), encoding a splicing factor aberrantly overexpressed in GBM [45] and NOP2/Sun RNA methyltransferase 6 (*NSUN6*) encoding the RNA 5-methyl cytosine (5mC) transferase that regulates the glioblastoma response to temozolomide [46]. Surprisingly, 33% of them (17/51) are genes mutated in children with neurodevelopmental disorders (NDDs; Appendix A), including the Activator of transcription and developmental regulator AUTS2 (*AUTS2*; MIM 607270) [47], SWI/SNF-related matrix-associated actin-dependent regulator of chromatin subfamily A member 5 (*SMARCA5*, MIM 603375) [48], KAT8 regulatory NSL complex subunit 1 (*KANSL1*; MIM 612452) [49], and Lysine methyltransferase 2A (*KMT2A;* MIM 159555) [50] that are NDD chromatin remodeling genes (KW:0991; Figure 4F; Appendix A) as *KDM5C* [29,31].

### 2.6. Analysis of Stem Cell-Associated Genes Expression in the GBM Cohorts

A growing body of evidence indicates that GBM may be generated from stem cell-like tumor cells (GSCs), sharing many properties with those of neural stem cells [51]. Our analysis of differentially methylated genes in the KDM5C^High^ and KDM5C^Low^ TCGA subgroups indicated “stem cell differentiation” as one of the top-ranked pathways (Reactome). Hence, to investigate the potential correlation between stem cell-derived genetic signatures and *KDM5C* expression, we measured—in our cohort of GBM patients—the expression levels of four GBM stemness markers: the transcription factor genes NANOG homeobox (*NANOG)*, organic cation/carnitine transporter 4 (*OCT4)*, and sex-determining region Y box 2 (*SOX2)* and the neuroectodermal marker *NESTIN* [52]. Interestingly, we observed a significant overexpression of *OCT4*, *SOX2*, and *NESTIN* in both tumor KDM5C subgroups compared to the control samples. Surprisingly, *NANOG* was largely overexpressed only in *KDM5C*^High^ patients compared both to *KDM5C*^Low^ and healthy groups (Figure 5A).

Searching for the potential correlation between the stemness and hypoxia markers, we found a significant positive trend between *SOX2* and *HIF1A* only in KDM5C^High^ patients (Figure 5B). Taken into consideration the clinical characteristics of GBM patients, we expanded our analysis by exploring the potential correlation with specific stemness marker profiling. Noteworthy, the two KDM5C subgroups showed significant changes in *NANOG* expression in the subset of patients without epilepsy (Appendix A) or with the wild-type *IDH1/2* genotype (Appendix A) or with *MGMT* methylation (Appendix A), while significant *OCT4* change was found in the subset of patients with the wild-type *IDH1/2* genotype (Appendix A). These results, together with the KDM5C-associated methylation profile of “stem cell differentiation” genes—observed in the larger and independent TCGA sample cohorts (Figure 4C)—corroborate the finding that GBM tumors are enriched in stem cell markers compared to the healthy tissue. These data shed light on a distinctive signature marked by a high co-expression of HIF1A-KDM5C-NANOG that could help to explain differences in the clinical phenotype.

Moving to 5-ALA FGS tissues, we detected a high level of *NANOG*, as well as of *SOX2* and *NESTIN* in the TC region compared with the TR and TM regions (Figure 5C). Based on this evidence, we therefore hypothesize that the three TMEs display distinctive stem cell-derived signatures that may depend on the heterogeneous differentiation stage of tumor cells whose further characterization could validate their prognostic values.

### 2.7. Expression Profiles of Tumor Progression Markers and Inflammatory Cytokines in GBM Samples

Since p53 expression has been postulated to be modulated by KDM5C (e.g., gastric cancer cell lines and gastric tumors) [10] and “p53 signaling” ranked among the top Reactome pathways enriched for differentially methylated genes between the KDM5C^High^ and KDM5C^Low^ subgroups (TCGA), we quantified p53 by immunohistochemistry in the biopsies of the entire cohort of tumor samples. The tumor samples were labeled as “strong”, “moderate”, “weak”, or “negative”, according to the p53 protein levels [53]. Interestingly, we found that the majority of the GBM samples having a “strong” p53 signal belonging to the KDM5C^High^ subgroup (Appendix A), in line with the literature data on gastric cancer [10]. Noteworthy, a strong immunoreactivity signal was found in most patients of the KDM5C^High^ subgroup with the earlier onset of GBM (median age at 48; Appendix A). However, unexpectedly, the only p53-negative patients were found in the same tumor subgroup (Appendix A). By the RT-PCR assay, an upregulation of the *TP53* transcript was found in the KDM5C^Low^ and KDM5C^High^ cohorts (Appendix A), as well as in CoCl_2_-treated T98G cells (Appendix A) and in the TC region of 5-ALA FGS tissues (Appendix A). Although it is still unknown, the molecular mechanism linking KDM5C to p53, our data highlight the expression of high levels of the p53 protein strongly associated with an adverse outcome in GBM in tumor samples presenting high levels of KDM5C.

Next, given the enrichment of genes differentially methylated belonging to the Reactome pathway “inflammation”, we measured the gene expression of Triggering Receptor Expressed on Myeloid cells-2 (*TREM2*), a microglia surface receptor involved in neuroinflammation that is overexpressed in the glioma and associated with tumor progression [54,55]. A pronounced increase of *TREM2* was detected in both GBM subgroups compared to the control tissues (Figure 6A). Furthermore, we measured the expression of the genes encoding proinflammatory cytokines Interleukin-6 (*IL6*), Interleukin-10 (*IL10*), and Interleukin-33 (*IL33*), associated with increased cell proliferation and tumorigenesis in GBM [56,57,58]. In line with the literature data [56,57,58], *IL10* and *IL33* were overexpressed in the entire cohort of GBM patients compared to the healthy samples (Figure 6B). However, *IL6* was robustly induced only in the *KDM5C*^High^ samples (Figure 6B), and interestingly, it showed a significant positive correlation with *NANOG* only in this tumor subtype (Figure 6C). Interestingly, exploring the clinical features of GBM patients, we found a significant change in *IL6* expression between the two *KDM5C* subgroups with epilepsy (Figure 6D). Moreover, measuring the *TREM2* expression and the inflammatory cytokine genes in the 5-ALA FGS tissues, we found that their levels are higher in TC areas compared to the TR and TM ones (Figure 6E,F), further corroborating the previous findings about the role of the inflammatory response in GBM. However, our data highlight how these molecular features of such a tumorigenesis-related process vary according to the surrounding TMEs in the context of GBM.

## 3. Discussion

The present work evaluated for the first time the association between alterations in the levels of the chromatin-oxygen sensor gene *KDM5C* and GBM pathogenesis. Based on the expression profile data in GBM tissues and the Cancer Genome Atlas (TCGA) database, we stratified GBM patients into two informative subtypes displaying high- and low-*KDM5C* levels. By an experimental approach mainly based on expression data, we revealed distinctive molecular profiles for hypoxia, stemness, and neuroinflammation markers. Although this somehow complicates the interpretation of our results, our study supports the hypothesis that diverse pathogenic mechanisms are implicated in the two subgroups.

Interestingly, we found a co-overexpression of *KDM5C* and the tumor prognostic genes *HIF1A*, *p75* and *survivin* (alias *BIRC5*), involved in hypoxia-mediated mechanisms, as aggressiveness and a poor therapeutic response. *HIF1A* encodes the oxygen-regulated transcription factor Hif-1α that controls a number of pathways involved in GBM aggressiveness [34], whose overexpression was observed in GBM patients with poor prognosis and low sensitivity to chemotherapy [37]. Very importantly, hypoxia-induced Hif-1α stabilization is mediated by increased levels of p75^NTR^, a marker of aggressiveness where the inhibition reduces migration, invasion, and stemness in GBM [39]. In turn, Hif-1α cooperates to regulate *survivin* expression, whose overexpression was negatively associated with the overall survival in GBM patients and positively associated with hypoxia and stemness [59]. Related to these previous findings, our data corroborate the strong interplay between p75^NTR^ and Hif-1α and between Hif-1α and Survivin in GBM and, above all, suggest that the p75^NTR^–HIF1A–KDM5C–survivin axis could be a new oncogenic hypoxia-mediated gene target with high therapeutic potential. Remarkably, as a consequence of KDM5C upregulation, a marked *BDNF* repression was found in the hypoxia condition. Since the hypoxia-mediated suppression of BDNF has been associated with neuronal loss and spatial memory impairment [60] and GBM patients frequently suffer cognitive deficits, including problems with attention and memory, we therefore conclude that BDNF decrease can contribute to the manifestation of these clinical signs in GBM patients. On the contrary, high BDNF acts on stem cells, promoting malignant progression [61].

It is important to highlight that hypoxia exerts pleiotropic effects in GBM as cell reprogramming towards a stem cell phenotype associated with high levels of stemness markers and activation of the inflammatory response with high levels of proinflammatory mediators [62]. According to previous findings, our study shows that the stemness genes *OCT4*, *SOX2*, and *NESTIN* are highly expressed in both the GBM sub-cohorts, but surprisingly, we observed that *NANOG* is overexpressed only in the *KDM5C*^High^ subgroup compared both to *KDM5C*^Low^ and healthy control. Since it has been reported that *NANOG* expression is controlled by Hif-1α and that they cooperate, promoting self-renewal in breast cancer stem cells [63], we therefore hypothesize that the GBM subgroup with HIF1A-KDM5C hyperactivity has a distinctive stemness expression signature. These findings could explain, at least in part, the peculiar intratumoral GBM heterogeneity, frequently characterized by a variability in stem-like cell composition, a feature that correlates with a varying degree of invasion capacity, resistance to conventional treatments, and tumor relapse.

Secondary inflammatory changes are frequently associated with tumor hypoxia by establishing a particular condition called “inflammatory hypoxia” [64]. Very importantly, our molecular analysis revealed high levels of *TREM2*, *IL10*, and *IL-33* in the KDM5C subgroups corroborating the critical role of these inflammatory mediators in GBM. However, only in KDM5C^High^ samples, we detected the *IL6* overexpression coupled with a positive correlation with the *NANOG* levels. These results further differentiate, from a molecular point of view, the two GBM subgroups, assigning a specific IL6-NANOG signature to the KDM5C^High^. Since IL6 overexpression is induced by hypoxia [65], and as IL6 may strongly induce *NANOG* expression and promote proliferation and stemness in other tumors [66,67,68], the co-expression of IL6-NANOG likely represents a hypoxia-induced module, which appears as a distinctive feature of the KDM5C^High^ GBM phenotype. Nevertheless, our results further emphasize TREM2 as a GBM biomarker, as recently reported [69].

By extending our analysis to tumor suppressor p53, whose expression associates with the pathological grade of glioma [70], a strong immunoreactivity signal was found in most patients of the KDM5C^High^ sub-cohort with an earlier onset of GBM. This finding may emphasize the promising use of p53 expression as an early predictive GBM marker in the KDM5C^High^ sub-cohort. Very importantly, the finding that GBM samples (TCGA) with high levels of *KDM5C* display DNA methylation changes mainly affecting genes involved in the hypoxia response, stem cell differentiation, inflammation, and p53 signaling further corroborates the role of KDM5C in GBM etiology and/or progression.

As proven by several reports, aberrant DNA methylation and/or histone modifications can disrupt the correct transcriptional control of clinically relevant genes and trigger malignant cellular transformation [71]. We therefore propose that, in an inflammatory-hypoxic microenvironment marked by high levels of HIF1A-KDM5C-IL6, tumor cells may dedifferentiate via a survivin-dependent pathway and acquire a particular stem-like phenotype marked by a high level of *NANOG* expression and associated, in turn, with high p53 levels (Figure 7A). Remarkably, taken into consideration the clinical features, the KDM5C^High^ subgroup showed an interesting correlation between the epilepsy outcome and high co-expression of *Ki67* and *IL6* (Figure 7A). Given the implication of KDM5C in NDDs with a seizure [26,29], we believe that the identification of these KDM5C-related signatures constitutes a step forward to understand the GBM-related epileptic phenotype. Indeed, Ki67 overexpression was proposed as a significant predictor of poor seizure control in GBM patients [72], while the activation of proinflammatory cytokines in response to inadequate homeostasis in the GBM tumor tissues was suggested as seizure susceptibility factors [73]. Since GBM-related seizures are often refractory to antiepileptic treatments, the characterization of these pathogenic players could open a new field of investigation for clinical treatment. Furthermore, other interesting correlations were found between *MGMT* methylation and high *NANOG* in KDM5C^High^ patients and, likewise, between the WT *IDH1/2* genotype with high p53 and low *NANOG* and *OCT4*. Although the functional relationships between these GBM tumor markers are not well-known in the literature, the identification of these gene expression profiles together with the analysis of the clinical features could contribute to shed light on the multiple pathogenic mechanisms involved and on the substantial heterogeneity of the GBM phenotype. On the other hand, the moniker ‘multiforme’ derives from the first histopathological description characterized by the presence of heterogeneous cell populations, where coexist functional subdomains within a single tumor mass [74]. Related to this consideration, our molecular study in the 5-ALA FGS tumor samples revealed a peculiar expression profile across the microenvironmental cells residing in different tumor niches with a higher expression of the stem cell markers in the tumor core and of *KDM5C* in the tumor rim. Approved in Europe and in the United States, the prodrug 5-ALA induces tumor fluorescence with high specificity and sensitivity across all histopathologic grades for malignant high-grade glioma (with a probability close to 100%) facilitating the sampling of the core region of the tumor, the infiltrating front, and the healthy tissue [75]. Machine learning tools applied to 5-ALA fluorescence data analysis combined with molecular profiles could further characterize the GBM phenotypes.

The association with somatic mutations in cancer-related genes remains to be explored in our GBM cohorts, both bulk and 5-ALA FGS tumors. Indeed, low *KDM5C* expression could be caused by mutations in *KDM5C* or in its positive regulatory genes, while mutations in negative regulatory genes could stimulate high *KDM5C* expression.

Another genetic phenomenon contributing to generate *KDM5C* expression variability is sex disparity. Since this gene is located on X chromosome and escapes X-inactivation (Xi escaper) in females [76] (Figure 7B), a gene dosing effect with two copies expressed in females and one copy expressed in males could contribute to this scenario. Interestingly, the GBM TCGA analysis showed a peculiar sex-related distribution with a female prevalence in the KDM5C^High^ vs. KDM5C^Low^ subgroups (Figure 7C). Further studies in larger patient collections can elucidate the functional relationship between the levels of this Xi escaper gene and sex disparity in GBM outcome and therapy. Even though the observed *KDM5C* expression variability could be caused by these factors, our study represents a significant step forward the understanding of the molecular determinants at the basis of GBM pathogenesis, contributing to better defining the stem cell- and inflammatory hypoxia-based stratification of GBM. Noteworthy, the discovery of specific inhibitors or modulators of *KDM5C* will prospect the identification of new GBM therapies with translational application in personalized treatments. In this regard, the discovery of potent histone demethylase inhibitors across the JMJ family abled to inhibit glioma cell proliferation, such as GSK-J1/J4, encourages the design of new small-molecule inhibitors with selective pharmacological intervention against *KDM5C* [77,78,79]. Finally, since KDM5C specifically regulates several hypoxia-related genes and targets hypoxia-mediated mechanisms are considered an attractive approach to improve the therapy outcome in GBM [34], the discovery of new hypoxia-related signatures could accelerate the identification of more effective mechanism-based therapies and the development of predictive biomarkers, potentially linked to tumor-related epilepsy.

## 4. Materials and Methods

Patients and clinical data. A total of 37 patients with histopathologically confirmed glioblastoma were included in this study. These patients, admitted to A.O.R.N. S. Anna and S. Sebastiano Hospital (Caserta, Italy), were 20 males and 17 females, ranging in age from 34 to 78 years (Table 1 and Appendix A). Tumor characteristics, including lesion sites; treatment strategy; molecular analysis of the tumor markers Ki-67, p53, *MGMT* methylation, and survival; and mortality, were retrospectively examined (Table 1 and Appendix A). Epilepsy was reported in 13 patients (35%). Patients received radiotherapy (75.6%) and chemotherapy (92%). Isocitrate dehydrogenase 1/2 (*IDH1/2*) mutation analysis was performed with 62.2% of patients showing IDH1/2 mutations compared to 37.8% with the wild-type allele (Table 1 and Appendix A). The study was reviewed and approved by the Ethics Committee and Institutional Review Board of A.O.R.N. S. Anna and S. Sebastiano Hospital.

Clinical Samples. Tumor tissues were collected during neurosurgery at A.O.R.N. S. Anna and S. Sebastiano Hospital (Caserta, Italy). Immediately after surgical excision, the tissues were immersed in liquid nitrogen and kept in liquid nitrogen until use. We obtained the written informed consent from each participant before surgery. The Research Ethics Committee of Campania Nord (San Giuseppe Moscati-Avellino) approved the study protocol (Prot. 359 on date 15/02/20). The study was implemented in compliance with the ethical guidelines of the 1975 Declaration of Helsinki.

Fluorescence-guided surgery (FGS) using 5-aminolevulinic acid (5-ALA). Each patient received an oral dose of 5-ALA (20 mg/kg body weight) 2 h prior to the induction of anesthesia. During the tumor resection, 5-ALA-induced PpIX fluorescence and autofluorescence were visualized using a commercially available fluorescence operative microscope (OPMI, Pentero 800, Carl Zeiss, Oberkochen, Germany). Total microscopic tumor resection was performed with this fluorescence-guided method, combined with a standard navigation system (Medtronic StealthStation S7 Surgical Navigation system). The visualized fluorescence intensity was classified as described elsewhere [44]. Specifically, three grades of fluorescence intensity were identified: (i) a charcoal red PpIX fluorescence defined as strong fluorescence, a lack of red PpIX fluorescence, defined as non-fluorescence, and a pink or orange fluorescence was defined as vague fluorescence. The phenomenon of vague PpIX fluorescence was postulated as deriving from a merging of charcoal red PpIX fluorescence with green autofluorescence, resulting in a change of the fluorescence color spectrum to orange (500-nm low-pass filter) or light red (420- and 450-nm low-pass filters).

Cell culture and plasmid transfection. Human glioblastoma cell-line T98G cells (GBM-derived T98G) were obtained from ATCC. Cells were maintained in Dulbecco’s modified Eagle’s medium (DMEM) (Gibco, Carlsbad, CA, USA) supplemented with 10% fetal bovine serum (FBS; Gibco, Carlsbad, CA, USA), penicillin (100 units/mL; Gibco, Carlsbad, CA, USA), and streptomycin (100 μg/mL; Gibco, Carlsbad, CA, USA) in a humidified 5% (*v/v*) CO_2_ atmosphere and were routinely tested for mycoplasma contamination. The wild-type (WT) human *KDM5C* cDNA cloned in the pcDNA3.1 vector (pcDNA_KDM5C) was a gift of Charles Schwartz (Greenwood Genetic Center, Greenwood, SC, USA). pcDNA3.1 empty and pcDNA3.1_KDM5C plasmids were purified with a plasmid DNA kit (Qiagen, Venlo, Limburg, Belgium). For transfection experiments, empty pcDNA3.1 and pcDNA3.1_*KDM5C* plasmids were transiently transfected into T98G cells using Lipofectamine 2000 reagent (Invitrogen, Waltham, MA, USA). T98G cells (1 × 10^6^) were seeded overnight, and the mixture of each plasmid DNA (8 μg) and Lipofectamine 2000 diluted in Opti-MEM was added to the cells, and the cells were cultured for 48 h. All experiments were performed at least three times. The primers used are listed in Appendix A.

Hypoxia treatment with cobalt chloride (CoCl_2_). Stock solution of 25 mM of cobalt chloride (CoCl_2_) was prepared in sterile distilled water and further diluted in the medium in order to obtain the final desired concentrations. T98G cells were treated with 50 μM CoCl_2_ and incubated for 24 h.

## Figures and Tables

**Figure 1 ijms-23-10250-f001:**
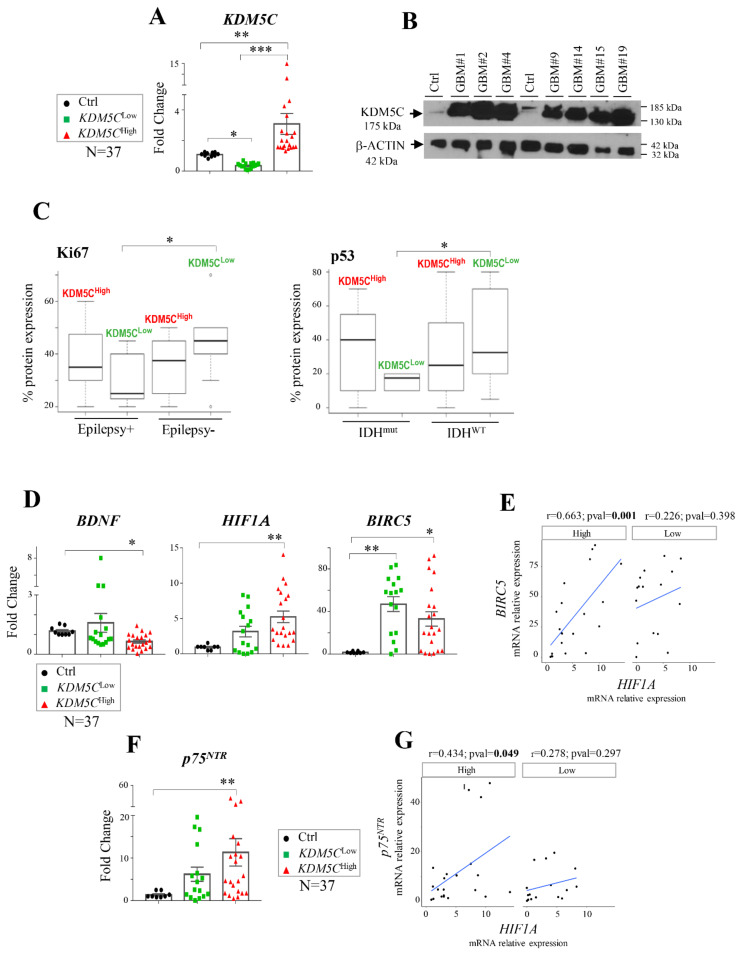
Expression analysis in KDM5C^Low^ and KDM5C^High^ cohorts. (**A**) Analysis of *KDM5C* expression by real-time PCR (RT-PCR) assay and (**B**) Western blotting. (**C**) Ki67 expression analysis in GBM patients with (Epilepsy+) or without epilepsy (Epilepsy−), and p53 expression analysis in GBM patients with wild-type (WT) or mutated *IDH1/2* (mut) alleles. (**D**) RT-PCR analysis of *BDNF*, *HIF1*A, and *BIR**C5*. (**E**) Scatter plots reporting the correlation—by linear regression analysis—of *HIF1A* and *BIRC5* expression in the KDM5C^Low^ and KDM5C^High^ subgroups. (**F**) RT-PCR analysis of *p75^NTR^* expression in healthy controls, KDM5C^Low^, and KDM5C^High^ GBM samples. (**G**) Scatter plot reporting the correlation—by linear regression analysis—of *HIF1A* and *p75^NTR^* expression in the KDM5C^Low^ and KDM5C^High^ subgroups. Each transcript analysis was performed in triplicate, and the samples were normalized with the TATA-Box Binding Protein (*TBP*) transcript. The bars indicate the mean ± SEM of repeated experiments. Asterisks indicate statistical significance compared with healthy control samples: *** *p* < 0.0005, ** *p* < 0.005, and * *p* < 0.05. The Western blotting experiment was repeated in triplicate, and the beta-actin antibody (β-ACTIN) was used as a loading control. In the scatter plots, the average expression value of healthy control samples was used as a reference. The Pearson’s correlation coefficient (r) and *p*-values (*p*) are indicated on the top of each plot. In the box plot, the black thick line inside the box represents the median of the values, and the 25th and 75th percentiles are shown as boundaries of the box. The Student’s *t*-test was applied with * *p* < 0.05.

**Figure 2 ijms-23-10250-f002:**
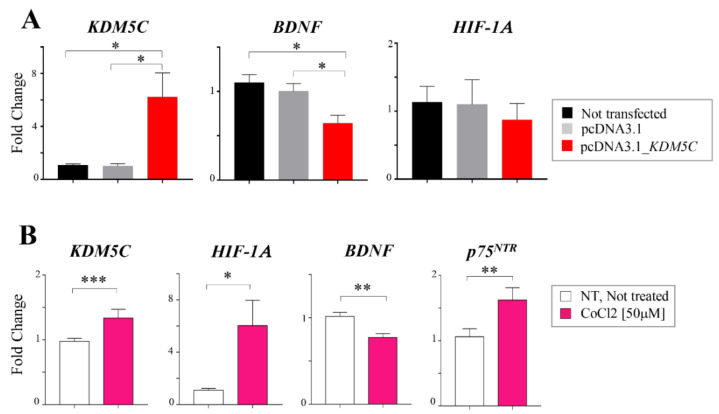
Expression analysis in T98G cell lines. (**A**) RT-PCR analysis of *KDM5C*, *BDNF*, and *HIF1A* transcripts in T98G cell lines transiently transfected with a plasmid expressing *KDM5C* or a plasmid containing the empty vector. (**B**) RT-PCR analysis of *KDM5C*, *HIF1A, BDNF*, and *p75^NTR^* in T98G cell lines exposed to cobalt chloride (CoCl_2_). Each transcript analysis was performed in triplicate, and the samples were normalized with the *18S* transcript. The bars indicate the mean ± dev.st. of repeated experiments. Asterisks indicate statistical significance compared with the control samples: *** *p* < 0.0005, ** *p* < 0.005, and * *p* < 0.05.

**Figure 3 ijms-23-10250-f003:**
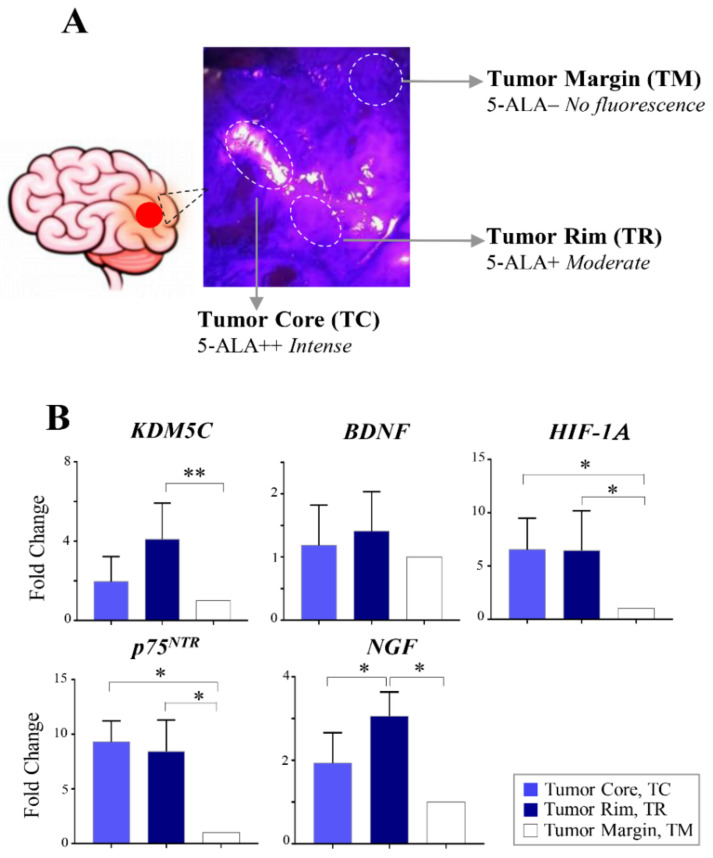
Expression analysis in GBM areas isolated by 5-aminolevulinic acid fluorescence-guided surgery. (**A**) Intraoperative photographs obtained in the GBM patient #45. 5-ALA administration showing the resection cavity illuminated with blue light, demonstrating intense and moderate 5-ALA fluorescence representing the tumor core and tumor rim, respectively. (**B**) RT-PCR analysis of the *KDM5C*, *BDNF*, *HIF1A, p75^NTR^*, and *NGF* transcripts in different layers identified by 5-ALA surgery. Each transcript analysis was performed in triplicate, and the samples were normalized with the TATA-Box Binding Protein (*TBP*) gene. The bars indicate the mean ± SEM of repeated experiments. Asterisks indicate statistical significance compared with the healthy control samples: ** *p* < 0.005 and * *p*< 0.05.

**Figure 4 ijms-23-10250-f004:**
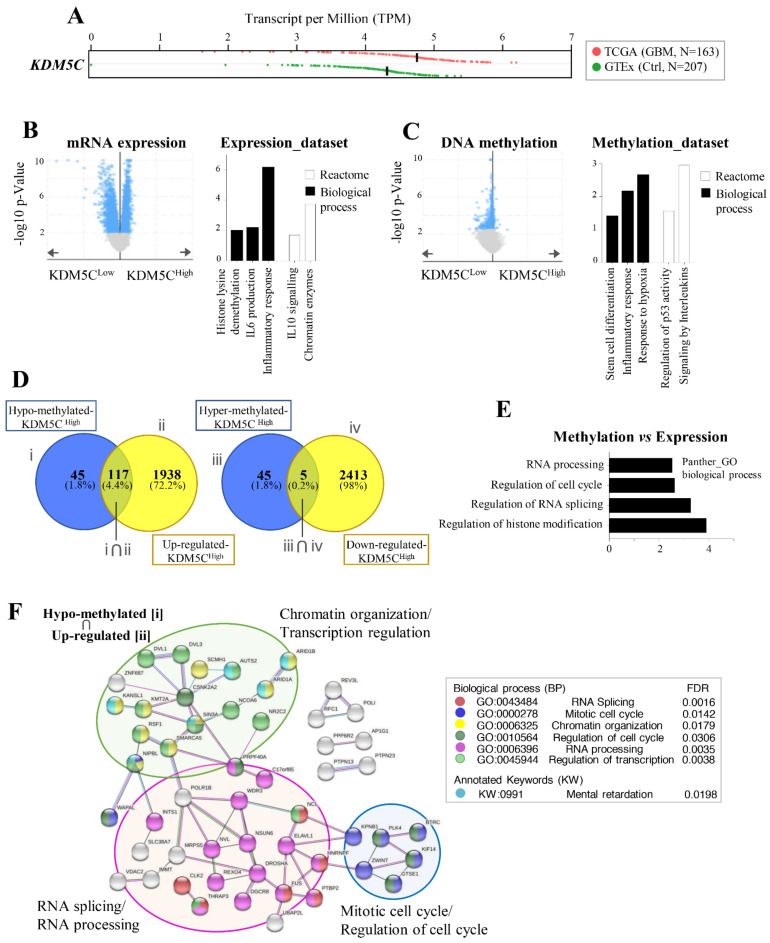
Features of the KDM5C^Low^ and KDM5C^High^ subgroups in the TCGA dataset. (**A**) *KDM5C* expression (transcript per million, RNA-Seq data) in the GBM samples (TCGA; N = 163) compared to control post-mortem brains (GTeX portal; N = 207). (**B**,**C**) On the left, volcano plots reporting differential mRNA expression and methylation between the two GBM subgroups (downloaded from cBioportal). On the right, bar graphs reporting the results of Gene Ontology and pathway analysis on differentially expressed (**B**) and methylated (**C**) genes between the two tumor subgroups. (**D**) Venn diagrams of hypomethylated/overexpressed genes and hypermethylated/downregulated genes in the KDM5C^High^ cohort. (**E**) Gene Ontology analysis of DEGs regulated by methylation changes (DEGs_meth) in KDM5C^High^ (KDM5C^High^/DEGs_meth) by using the Panther classification system (www.pantherdb.org; accessed on 1 June 2022). (**F**) Network analysis of protein–protein interactions for KDM5C^High^/DEGs_meth genes using the STRING database.

**Figure 5 ijms-23-10250-f005:**
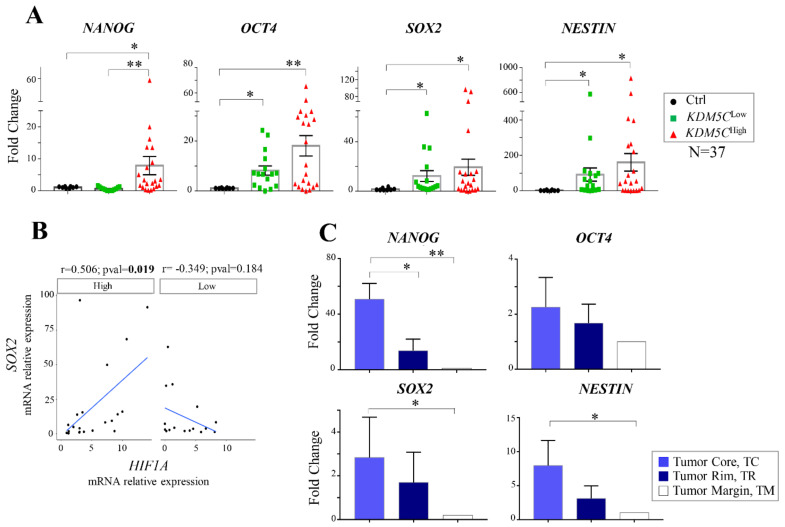
Expression analysis of the stem cell-associated genes in GBM tumor samples. (**A**) RT-PCR analysis of *NANOG*, *OCT4*, *SOX2*, and *NESTIN* in the KDM5C^Low^ and KDM5C^High^ cohorts. (**B**) Scatter plot reporting the correlation—by linear regression analysis—of *HIF1A* and *SOX2* in the KDM5C^Low^ and KDM5C^High^ subgroups. The average expression value of healthy control samples was used as a reference. The Pearson’s correlation coefficient (r) and *p*-values (*p*) are indicated on the top of each plot. (**C**) RT-PCR analysis of *NANOG*, *OCT4*, *SOX2*, and *NESTIN* in 5-ALA FGS tissues. Each transcript analysis was performed in triplicate, and the samples were normalized with the TATA-Box Binding Protein (*TBP*) gene. The bars indicate the mean ± SEM of repeated experiments. Asterisks indicate statistical significance compared with healthy control samples: ** *p* < 0.005 and * *p*< 0.05.

**Figure 6 ijms-23-10250-f006:**
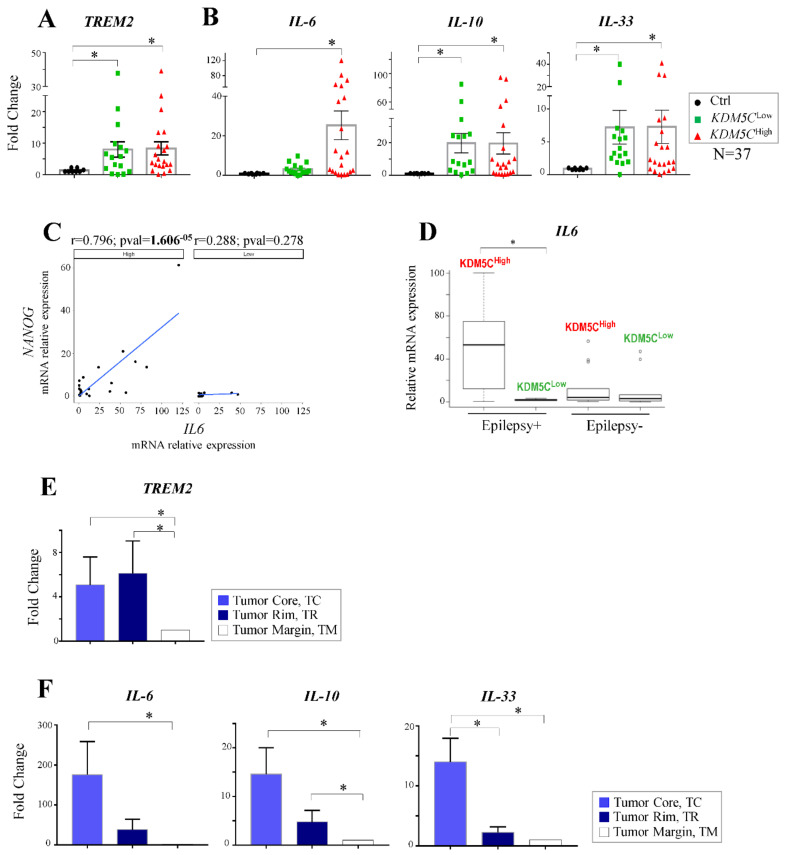
Expression profiles *of TREM2* and proinflammatory cytokine genes in GBM tumor samples. (**A**,**B**) RT-PCR assay in the KDM5C^Low^ and KDM5C^High^ subgroups. (**C**) Scatter plot reporting the correlation—by linear regression analysis—of *NANOG* and *IL6* in the KDM5C^Low^ and KDM5C^High^ subgroups. (**D**) *IL6* expression analysis in GBM patients with (Epilepsy+) or without epilepsy (Epilepsy−). (**E**,**F**) RT-PCR assay in 5-ALA FGS tissues. Each transcript analysis was performed in triplicate, and the samples were normalized with the *TBP* gene. The bars indicate the mean ± SEM of repeated experiments. Asterisks indicate statistical significance compared with the healthy control samples: * *p* < 0.05. In the scatter plots, the average expression value of the healthy control samples was used as a reference. The Pearson’s correlation coefficient (r) and *p*-values (*p*) are indicated on the top of each plot.

**Figure 7 ijms-23-10250-f007:**
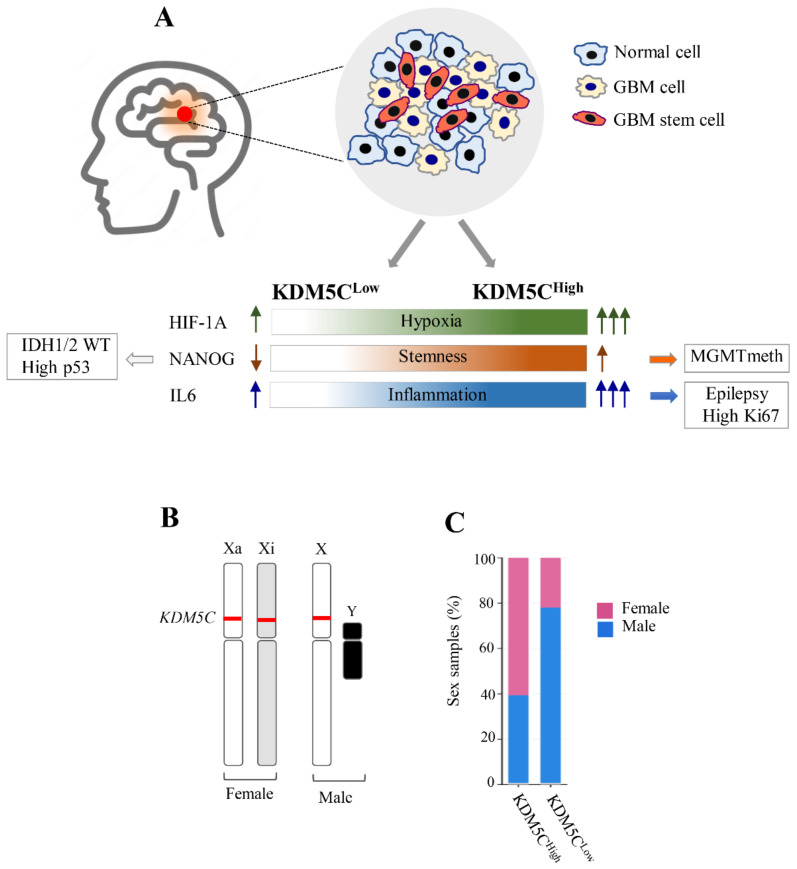
Molecular features of GBM patients stratified by expression data of the chromatin oxygen sensor gene *KDM5C*. (**A**) High expression levels of the tumor markers *HIF1A*, *NANOG*, and *IL6* are associated with a high level of *KDM5C*. (**B**) *KDM5C* expression variability in female and male individuals. Since *KDM5C* escapes X-inactivation, two active alleles (in red) are present in female individuals, unlike the male individuals, in which only one allele drives the expression of *KDM5C*. (**C**) Sex-related distribution in the KDM5C^High^ and KDM5C^Low^ GBM subgroups identified in TCGA. Xa, X chromosome active; Xi, X chromosome inactive.

**Table 1 ijms-23-10250-t001:** Demographic data, tumor characteristics, and treatments strategies of 37 GBM patients enrolled in this study.

Clinicopathological Parameters		No. of Patients (Frequency)
Gender	Male	20 (54%)
	Female	17 (46%)
Epilepsy	Yes	13 (35%)
	No	24 (65%)
Location of lesion	Parietal	4 (10.8 %)
	Frontal	7 (18.9%)
	Temporal	5 (13.6%)
	Rolandic	3 (8.1%)
	Corpus Callosum	2 (5.4%)
	Frontotemporal	4 (10.8%)
	Temporo-Parietal	3 (8.1%)
	Frontoparietal	3 (8.1%)
	Parieto-Occipital	2 (5.4%)
	Others	4 (10.8%)
Lobe localization	Right hemisphere	17 (45.9%)
	Left hemisphere	17 (45.9%)
	Others	3 (8.2%)
Lesion number	Single	33 (89%)
	Multifocal	4 (11%)
Surgical approach	Total resection with standard method	25 (68%)
	Total resection with 5-ALA	9 (24%)
	Biopsy	3 (8%)
Radiotherapy	Yes	28 (75.6%)
	No	9 (24.4%)
Chemotherapy	Yes	34 (92%)
	No	3 (8%)
*MGMT* methylation	Yes	33 (89.2%)
	No	4 (10.8%)
*IDH1/2* mutation	WT	14 (37.8%)
	MUT	23 (62.2%)
p53 expression	Yes	32 (86.5%)
	No	5 (13.5%)
Ki67 expression	Yes	37 (100%)
Survival	--	17 (46%)
Mortality rate	--	20 (54%)

## Data Availability

Publicly archived datasets analyzed: The Genotype Tissue Expression database (GTEx; https://gtexportal.org/home/; accessed on 10 March 2022). The Cancer Genome Atlas (TCGA) Consortium was retrieved from public repository cBioportal (https://www.cbioportal.org; accessed on 10 March 2022).

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
