# Peer review of "The Chromatin-Oxygen Sensor Gene KDM5C Associates with Novel Hypoxia-Related Signatures in Glioblastoma Multiforme"

_ijms, 2022, doi:10.3390/ijms231810250_

Round 1

Reviewer 1 Report

The manuscript describes for the first time the contribution of KDM5C in GBM by analyzing clinical, expression and functional data as well as publicly available omic datasets. 

Overall, the manuscript is well written and structured and the results shown are well fundamented. However, in the test the authors mentioned that "KDM5C has a role in the tumor cell proliferation, migration, and drug resistance". However, authors haven't done any functional experiments regarding these functions in T98G cell lines. Authors could include some experiments of T98G cell line (not transfected, pcDNA3.1 and pcDNA3.1_KDM5C) such as proliferation assay, oncospheres formation assay as well as migration and invasion assays, in order to corroborote their hypothesis. 

Moreover, I would like to thank again authors because the manuscript is well structured and is very easy to follow.

Author Response

Reviewer 1 Comments

The manuscript describes for the first time the contribution of KDM5C in GBM by analyzing clinical, expression and functional data as well as publicly available omic datasets.

Overall, the manuscript is well written and structured and the results shown are well fundamented. Point 1. However, in the test the authors mentioned that "KDM5C has a role in the tumor cell proliferation, migration, and drug resistance".

However, authors haven't done any functional experiments regarding these functions in T98G cell lines. Authors could include some experiments of T98G cell line (not transfected, pcDNA3.1 and pcDNA3.1_KDM5C) such as proliferation assay, oncospheres formation assay as well as migration and invasion assays, in order to corroborate their hypothesis.

Response 1. We would like to thank the Reviewer 1 giving the opportunity to clarify this point.

As mentioned in the Introduction section at line 63-64, the study of Xu et al. indicates that in gastric cancer cells KDM5C has a role in the tumor cell proliferation, migration, and drug resistance. We agree with the referee’s suggestion that it would be very interesting to verify if KDM5C has a similar role in GBM but the analysis of these functions is outside the scope of this study. The experiments suggested including proliferation assay, oncospheres formation assay, migration and invasion assays could become the next step of our project on GBM.

Point 2. Moreover, I would like to thank again authors because the manuscript is well structured and is very easy to follow.

Response 2. Thank you so much for this comment. I really appreciate.

1.0.0.20

Reviewer 2 Report

In this study, the authors looked at the contribution of the chromatin remodeling gene Lysine (K)-specific demethylase 5C (KDM5C) in GBM by conducting functional and expression analyses with cell lines and patient GBM tumor samples (n=37). These data were further integrated with publicly available omics datasets, which allowed for additional correlations to be made. Overall, the authors show that there is a correlation at transcript level between KDM5C expression and prognostic markers such as HIF1ASurvivin, and p75, which establishes KDM5C as a potential therapeutic target in the hypoxic microenvironment in GBM. One innovative aspect of their study was the use of 5-ALA fluorescence-guide surgery (FGS) for a spatial analysis of patient GBM tumors which allowed to inform on KDM5C expression status in different regions of each tumor (i.e., tumor core versus tumor rim versus tumor margin).  By employing this strategy, the authors found higher levels of KDM5C and HIF1A transcripts in the tumor core and rim compared to the tumor margins. Furthermore, by analyzing TCGA gene expression and methylation data grouped according to KDM5C expression, the authors found that high levels of stem cell (e.g., NANOG, etc.) and inflammatory (e.g., IL6, etc.) transcripts were differentially expressed in KDM5Chigh versus KDM5Clow subgroups.  

While I find the study very interesting, the vast majority of the data presented in this study are expression data with very little evidence provided at protein level for the various KDEM5C molecular partners analyzed in the study. This obviously complicates the interpretation of the data and represents a limitation of the study, which needs to be properly acknowledged by the authors. Furthermore, I have a number of comments and questions for the authors.     

1.   Table 1 shows that 23 out of 37 GBM patients had IDH1/2 mutated tumors. This suggests that these patients were diagnosed with secondary glioblastomas. Therefore, it would be interesting for the potential reader to see a breakdown of these clinical data based on GBM type (i.e., primary versus secondary and newly diagnosed versus recurrent) and whether these could be further stratified based on KDM5C expression levels. 

2.   BDNF—a negative KDM5C target—is used a readout for KDM5C overexpression. Assuming that BDNF is downregulated only in those GBM regions where KDM5C is overexpressed, what is the pathological significance of this downregulation, if it is known? In other words, perhaps the authors could clarify whether BDNF had been reported to be involved (and how) in the progression of GBM.     

3.     In the studies done with T98G cells, it would be particularly important to show the impact of KDM5Coverexpression on Hif-1α stability and retention at protein level. The lines 225-226 read “Interestingly, Hif-1α increase correlates with high levels of KDM5C mRNA (Fig. 2B).” However, the increases in HIF-1A are shown at transcript level only. Since this critical oxygen sensor is regulated at protein level, any changes in Hif-1α need to be validated at protein level.  Moreover, the conclusion from lines 229-231 which reads “Overall, in vitro data on a GBM tumor cell line indicate that in hypoxic conditions, p75NTRinduces HIF-1α stabilization that in turn induces KDM5C, a negative regulator of BDNF.” is not properly supported by the presented data. A Western blot analysis in T98G cells is currently missing in support of such conclusion.   

4.     Similarly, in the expression studies done with patient tumoral tissue stained with 5-ALA, complementary immunohistochemistry (IHC) analyses would be needed (i.e., staining for KDM5C, BDNF, Hif-1α, p75NTR and NGF) to investigate and demonstrate whether these proteins are indeed differentially regulated in the three different regions of the tumor (i.e., tumor core versus tumor rim versus tumor margin).  

5.   Lastly, perhaps the authors could further comment in the Discussion section about the translational implications of their findings. Specifically, if there are any drugs (either approved or in development) known to disrupt/inhibit KDM5C or its binding partners. 

Author Response

Reviewer 2 Comments

In this study, the authors looked at the contribution of the chromatin remodeling gene Lysine (K)-specific demethylase 5C (KDM5C) in GBM by conducting functional and expression analyses with cell lines and patient GBM tumor samples (n=37). These data were further integrated with publicly available omics datasets, which allowed for additional correlations to be made.

Overall, the authors show that there is a correlation at transcript level between KDM5C expression and prognostic markers such as HIF1A, Survivin, and p75, which establishes KDM5C as a potential therapeutic target in the hypoxic microenvironment in GBM. One innovative aspect of their study was the use of 5-ALA fluorescence-guide surgery (FGS) for a spatial analysis of patient GBM tumors which allowed to inform on KDM5C expression status in different regions of each tumor (i.e., tumor core versus tumor rim versus tumor margin). By employing this strategy, the authors found higher levels of KDM5C and HIF1A transcripts in the tumor core and rim compared to the tumor margins. Furthermore, by analyzing TCGA gene expression and methylation data grouped according to KDM5C expression, the authors found that high levels of stem cell (e.g., NANOG, etc.) and inflammatory (e.g., IL6, etc.) transcripts were differentially expressed in KDM5Chigh versus KDM5Clow subgroups.

Point A. While I find the study very interesting, the vast majority of the data presented in this study are expression data with very little evidence provided at protein level for the various KDEM5C molecular partners analyzed in the study. This obviously complicates the interpretation of the data and represents a limitation of the study, which needs to be properly acknowledged by the authors.

Response A. Thank you very much for this comment. Even though our approach -mainly based on expression data- could be considered a limitation, our results highlight novel molecular features of GBM linking the levels of the oxygen sensor KDM5C to the expression levels of other molecular players. On the other hand, giving the pleiotropic role of KDM5C as transcriptional regulator, we cannot exclude that this chromatin remodelling enzyme takes part to the expression control of these molecules.

However, to better clarify this point, we modified in the Discussion section the sentence at page 15 line 440,

“By an experimental approach mainly based on expression data, we revealed distinctive molecular profiles for hypoxia, stemness and neuroinflammation markers. Although this somehow complicates the interpretation of our results, our study supports the hypothesis that diverse pathogenic mechanisms are implicated in the two GBM subgroups.”

Furthermore, I have a number of comments and questions for the authors.    

Point 1. Table 1 shows that 23 out of 37 GBM patients had IDH1/2 mutated tumors. This suggests that these patients were diagnosed with secondary glioblastomas. Therefore, it would be interesting for the potential reader to see a breakdown of these clinical data based on GBM type (i.e., primary versus secondary and newly diagnosed versus recurrent) and whether these could be further stratified based on KDM5C expression levels.

Response 1. Thank you for this interesting suggestion. About the stratification of primary versus secondary, we did not find any correlation with the levels of KDM5C expression. Because recurrent GBM cases are absent in our cohort, it was impossible to verify a potential stratification of newly diagnosed versus recurrent GBM.

Point 2. BDNF—a negative KDM5C target—is used a readout for KDM5C overexpression. Assuming that BDNF is downregulated only in those GBM regions where KDM5C is overexpressed, what is the pathological significance of this downregulation, if it is known? In other words, perhaps the authors could clarify whether BDNF had been reported to be involved (and how) in the progression of GBM.

Response 2. We apologize for the insufficient comment on BDNF results. To overcame that, we added the following sentence at page 15 line 458:

“Remarkably, as consequence of KDM5C upregulation, a marked BDNF repression was found in hypoxia-condition. Since hypoxia-mediated suppression of BDNF has been associated with neuronal loss and spatial memory impairment [60], and that GBM patients frequently suffer of cognitive deficits including problems with attention and memory, we therefore conclude that BDNF decrease can contribute to the manifestation of these clinical signs in GBM patients. On the contrary, high BDNF acts on stem cells promoting malignant progression [61]”.

We therefore added the following references changing the numbering of the references in the manuscript and in the supplementary data.

  1. Kumar R, Jain V, Kushwah N, Dheer A, Mishra KP, Prasad D, Singh SB. Role of DNA Methylation in Hypobaric Hypoxia-Induced Neurodegeneration and Spatial Memory Impairment. Ann Neurosci. 2018 Dec;25(4):191-200. doi: 10.1159/000490368.

  1. Muthukrishnan SD, Alvarado AG, Kornblum HI. Building Bonds: Cancer Stem Cells Depend on Their Progeny to Drive Tumor Progression. Cell Stem Cell. 2018 Apr 5;22(4):473-474. doi: 10.1016/j.stem.2018.03.008. PMID: 29625062; PMCID: PMC8320684.

Point 3. In the studies done with T98G cells, it would be particularly important to show the impact of KDM5Coverexpression on Hif-1α stability and retention at protein level. The lines 225-226 read “Interestingly, Hif-1α increase correlates with high levels of KDM5C mRNA (Fig. 2B).” However, the increases in HIF-1A are shown at transcript level only. Since this critical oxygen sensor is regulated at protein level, any changes in Hif-1α need to be validated at protein level. Moreover, the conclusion from lines 229-231 which reads “Overall, in vitro data on a GBM tumor cell line indicate that in hypoxic conditions, p75NTRinduces HIF-1α stabilization that in turn induces KDM5C, a negative regulator of BDNF.” is not properly supported by the presented data. A Western blot analysis in T98G cells is currently missing in support of such conclusion.

Response 3. We thank the Reviewer 2 for these comments and we agree with her/him that the description of this part needs to be improved. However, we wish to underline that for technical problems due to the restricted quantity of tissue samples, we limited our analysis by testing only the mRNA levels of HIF1A. Given that, following her/his comment, we modified the sentences mentioned:

At lines 223-225 read “Interestingly, HIF1A mRNA increase correlates with high levels of KDM5C mRNA (Fig. 2B).”

At lines 228-230 “Overall, in vitro data on a GBM tumor cell line indicate that in hypoxic conditions, high levels of p75NTR may correlate with high levels of HIF1A and of KDM5C, a negative regulator of BDNF”.

Point 4. Similarly, in the expression studies done with patient tumoral tissue stained with 5-ALA, complementary immunohistochemistry (IHC) analyses would be needed (i.e., staining for KDM5C, BDNF, Hif-1α, p75NTR and NGF) to investigate and demonstrate whether these proteins are indeed differentially regulated in the three different regions of the tumor (i.e., tumor core versus tumor rim versus tumor margin).

Response 4. We thank the Reviewer 2 for this comment and we agree with her/him that complementary immunohistochemistry (IHC) analysis represents an important tool to validate 5-ALA data. However, giving the limited quantity of each sample it was impossible for us to carry out this analysis.

Point 5. Lastly, perhaps the authors could further comment in the Discussion section about the translational implications of their findings. Specifically, if there are any drugs (either approved or in development) known to disrupt/inhibit KDM5C or its binding partners.

Response 5. Thank you so much for this very interesting suggestion. We added the following sentence at line 550:

“Noteworthy, the discovery of specific inhibitors or modulators of KDM5C will prospect the identification of new GBM therapies with translational application in personalized treatments. In this regard, the discovery of potent histone demethylase inhibitors across the JMJ family abled to inhibit glioma cell proliferation, as GSK-J1/J4, encourages the design of new small-molecule inhibitors with selective pharmacological intervention against KDM5C [77-79]”.

We therefore added the following references changing the numbering of the references in the manuscript and in the supplementary data.

  1. Kruidenier, L., Chung, Cw., Cheng, Z. et al. A selective jumonji H3K27 demethylase inhibitor modulates the proinflammatory macrophage response. Nature 488, 404–408 (2012). https://doi.org/10.1038/nature11262

  1. Hashizume R, Andor N, Ihara Y, Lerner R, Gan H, Chen X, Fang D, Huang X, Tom MW, Ngo V, Solomon D, Mueller S, Paris PL, Zhang Z, Petritsch C, Gupta N, Waldman TA, James CD. Pharmacologic inhibition of histone demethylation as a therapy for pediatric brainstem glioma. Nat Med. 2014 Dec;20(12):1394-6. doi: 10.1038/nm.3716.

  1. Sui A, Xu Y, Li Y, Hu Q, Wang Z, Zhang H, Yang J, Guo X, Zhao W. The pharmacological role of histone demethylase JMJD3 inhibitor GSK-J4 on glioma cells. Oncotarget. 2017 Aug 2;8(40):68591-68598. doi: 10.18632/oncotarget.19793. PMID: 28978140; PMCID: PMC5620280.
1.0.0.20

Round 2

Reviewer 1 Report

Thank you for the response. 

Thus, I consider accepted the manuscript

Reviewer 2 Report

I would like to thank the authors for addressing my comments and for the further clarifications that were added to the text of their manuscript. As a general observation, even though it was not feasible to generate complementary IHC data due the absence of enough patient tissue samples, confirmatory ICC data could still be generated using the readily available glioma cell lines. Nonetheless, the improved Discussion section which now includes additional clarifications added by the authors will certainly benefit any potential readers.